# Structurally heterogeneous ribosomes cooperate in protein synthesis in bacterial cells

Karla Helena-Bueno[1,6], Sophie Kopetschke [2,6], Sebastian Filbeck [2], Lewis I. Chan[1], Sonia Birsan [1], Arnaud Baslé[1], Maisie Hudson [1], Stefan Pfeffer [2] ✉, Chris H. Hill [3,4,5] ✉ & Sergey V. Melnikov [1] ✉

Ribosome heterogeneity is a paradigm in biology, pertaining to the existence of structurally distinct populations of ribosomes within a single organism or cell. This concept suggests that structurally distinct pools of ribosomes have different functional properties and may be used to translate specific mRNAs. However, it is unknown to what extent structural heterogeneity reflects genuine functional specialization rather than stochastic variations in ribosome assembly. Here, we address this question by combining cryo-electron microscopy and tomography to observe individual structurally heterogeneous ribosomes in bacterial cells. We show that 70% of ribosomes in *Psychrobacter urativorans* contain a second copy of the ribosomal protein bS20 at a previously unknown binding site on the large ribosomal subunit. We then determine that this second bS20 copy appears to be functionally neutral. This demonstrates that ribosome heterogeneity does not necessarily lead to functional specialization, even when it involves significant variations such as the presence or absence of a ribosomal protein. Instead, we show that heterogeneous ribosomes can cooperate in general protein synthesis rather than specialize in translating discrete populations of mRNA.

Over five decades ago, researchers discovered that cells have the remarkable capacity to modify the molecular composition of their ribosomes[1–3]. This occurs through the expression of paralogous ribosomal proteins and rRNA[2–11], chemical modifications of these ribosome components[12–15], or variations in the stoichiometry of ribosomal proteins[2,3,16–18]. As a result, organisms ranging from bacteria to humans can express structurally distinct ribosomes.

These findings have evoked the idea that structurally distinct pools of ribosomes have different functional properties and may be used in a functionally specialised manner to translate specific mRNAs[5]. Over the years, this idea was supported by many correlative studies. For instance, some types of ribosome heterogeneity were found to accompany normal progression of cells and organisms through stages of lifecycle[6,19]. Others occur conditionally, in response to environmental factors, including starvation[8,9], antibiotic treatments[4], or viral infection[10,11], suggesting adaptive functions. Furthermore, ribosome profiling analyses revealed certain structurally distinct ribosomes may have uneven distribution on cellular mRNAs[20]. Together, these findings have led to the increasingly popular hypothesis that cells can express distinct populations of autonomous, functionally specialized ribosomes that fulfil dissimilar activities in protein synthesis[19–23].

However, other studies support an alternative hypothesis: that many types of ribosome heterogeneity are stochastic and functionally neutral, or are related to control of ribosome biogenesis[24–26]. Structural

[1]Biosciences Institute, Newcastle University, Newcastle upon Tyne, UK. [2]Centre for Molecular Biology, Heidelberg University, Heidelberg, Germany. [3]York Structural Biology Laboratory, University of York, York, UK. [4]York Biomedical Research Institute, University of York, York, UK. [5]Department of Biology, University of York, York, UK. [6]These authors contributed equally: Karla Helena-Bueno, Sophie Kopetschke. ✉e-mail: s.pfeffer@zmbh.uni-heidelberg.de; chris.hill@york.ac.uk; sergey.melnikov@ncl.ac.uk

studies have revealed that most variations in paralogous rRNA or ribosomal proteins are located outside the active centres of the ribosome, excluding these variations from having a direct impact on protein synthesis. Furthermore, evolutionary analyses indicate that variations in ribosomes within the same species are subtle compared to variations between different species that share the core mechanisms of protein synthesis[27,28]. Additionally, certain paralogs of rRNA and ribosomal proteins are mutually interchangeable and dispensable for cell function. For example, the parasitic eukaryote *Plasmodium falciparum* was shown to produce dissimilar rRNA at different developmental stages, including A-type rRNA during the asexual phases within vertebrate hosts and S-type rRNA during the sporozoite stage in the mosquito vector, which encouraged the idea of specialized ribosomes[6]. However, subsequent genetic experiments showed that when *Plasmodium* parasites lack individual isoforms of rRNA genes, they are able to complete development in both the vertebrate and mosquito hosts, demonstrating the redundant nature of these rRNA isoforms[29].

Thus, despite unambiguous evidence for ribosome heterogeneity, the field remains controversial[30], largely due to the limitations of current tools in observing individual and structurally dissimilar ribosomes during protein synthesis in vivo. Consequently, the following key questions remain unanswered: Do heterogeneous ribosomes differ in their core activities during protein synthesis? And if so, can organisms segregate structurally dissimilar ribosomes to enable their specialized functions within single cells?

In recent years, advances in cryo-electron tomography (cryo-ET) have made it possible to directly observe the distribution, composition, and activity of individual ribosomes within the natural environment of a cell[31]. Here, we use this powerful technology to directly visualise structurally heterogeneous ribosomes during protein synthesis within single bacterial cells.

In this work, we identify and investigate the ribosome heterogeneity generated by the ribosomal protein bS20. This protein, discovered in *Escherichia coli* in the 1970s, was initially thought to represent two different proteins, S20 in the small ribosomal subunit and L26 in the large ribosomal subunit[32]. Later, S20 and L26 were shown to be identical, and renamed as protein S20/L26[33]. Since then, S20/L26 has been repeatedly observed in both the large and the small ribosomal subunits isolated from *E. coli* cells[34–39] or assembled in vitro[40]. However, when ribosome structures were determined through X-ray crystallography, S20/L26 was found only in the small ribosomal subunit of *E. coli* and other bacteria[41–43]. Hence, despite the biochemical evidence, the apparent presence of bS20 in the large subunit was annotated as an artifact of ribosome isolation[44–47]. Since then, studies of bacterial ribosomes were conducted with the assumption that this protein binds only a single site in the ribosome[45–47]. However, using cryo-EM and cellular cryo-ET analysis of ribosomes from the bacterium *Psychrobacter urativorans*, we show here that protein bS20 in the large ribosomal subunit is not an artifact of ribosome isolation but an actual component of a subset of ribosomes in bacterial cells.

## Results

### bS20 is a structural component of the large ribosomal subunit
We identified heterogeneous ribosomes as an unexpected outcome of our studies of the ribosomal stress response in *P. urativorans*, the cold-adapted γ-proteobacterium typically found in Arctic ornithogenic soil[48]. During cryo-EM analysis of *P. urativorans* ribosomes isolated from cold-shocked bacteria (Supplementary Figs. 1 and 2), we observed an additional, previously overlooked small protein on the solvent side of the large subunit, in the vicinity of protein uL4 (Fig. 1a, b). To identify this protein, we revisited our mass-spectrometry analyses of the isolated 70S *P. urativorans* ribosomes but did not find any additional ribosome-binding proteins, besides previously described ribosome partners[49]. Therefore, we took an alternative approach by manually building the backbone and using

FoldSeek[50] to identify proteins with similar predicted structures. Independently, we used ModelAngelo[51] to automatically build a model de-novo for the unknown protein. Our FoldSeek search revealed that the unknown protein showed significant similarity to only a single protein in the *P. urativorans* proteome, specifically to ribosomal protein bS20 (Supplementary Fig. 3, Supplementary Table 1). Consistent with this designation, ModelAngelo predicted a primary sequence highly similar to bS20 for the most well-resolved segments of the unknown density (Supplementary Fig. 4). Our subsequent docking of the bS20 structure revealed that, compared to the previously identified bS20 molecule in the small subunit, the newly identified bS20 molecule in the large subunit has a disordered N-terminus (residues 1-18), but the rest of the structure is nearly identical, as evidenced by a Cα-atom R.M.S.D. of 0.97 Å. Collectively, these analyses show that protein bS20 can simultaneously bind to two distinct sites in the 70S ribosome (Fig. 1a).

To test whether bS20 binds to the large subunit of ribosomes constantly or only in response to stress, and to exclude its binding as an artifact of ribosome isolation, we determined the structure of *P. urativorans* ribosomes using cryo-ET of non-stressed, exponentially growing *P. urativorans* cells (Supplementary Fig. 5) thinned by cryo-Focused Ion Beam (FIB) milling. Our classification of active states confirmed that virtually all ribosomes were associated with a P-site tRNA and translation factors instead of hibernation factors like Balon or RaiA, consistent with active translation in our cryo-ET samples. This analysis confirmed the presence of an additional copy of bS20 in the structure of the large subunit of ribosomes observed inside the bacterial cells (Fig. 1c, d; Supplementary Fig. 6).

### An additional copy of bS20 is present only in a subset of cellular ribosomes
Our cryo-ET maps of ribosomes visualised within exponentially growing bacterial cells indicated that the bS20 signal in the large subunit was weaker compared to other ribosomal proteins, suggesting substoichiometric binding. Therefore, we estimated bS20 stoichiometry at both of its binding sites by performing focused classification. Our analysis showed that bS20 is present in 100% of the small subunits but only in approximately 67% of large ribosomal subunits. This occupancy is consistent between different cells (standard deviation: 3%) (Fig. 2a, b; Supplementary Fig. 6, Supplementary Fig. 7). We also performed a similar analysis on our cryo-EM dataset of purified *P. urativorans* ribosomes. This revealed similar proportions of bS20 (~77%) bound to the large subunit, indicating that this binding is biochemically stable and likely independent of the growth conditions, because similar levels of bS20 were detected in ribosomes from cold-shocked and actively growing cells (Supplementary Fig. 1). Thus, we found that *P. urativorans* cells contain two populations of structurally distinct ribosomes, with one population carrying one copy of bS20 (1xbS20), and the other population carrying two copies of bS20 (2xbS20).

### Two types of ribosomes are used interchangeably and cooperate in protein synthesis
We next asked whether exponentially growing bacterial cells use these structurally heterogeneous ribosomes interchangeably or in a manner consistent with functional specialization. To answer this, we assessed several key functional characteristics of ribosomes in situ, examining the structure and behaviour of ribosomes in the cytosol of *P. urativorans* cells using molecular-resolution cryo-ET. We first measured ribosome distribution between functional states throughout the translation cycle, including aminoacyl-tRNA binding, and pre- and post-translocation stages of elongation. This analysis revealed that 1xbS20 and 2xbS20 ribosomes have the same proportions of each functional state (Fig. 3a; Supplementary Figs. 6 and 7). We then measured ribosome association with Trigger Factor to assess the in vivo binding of ribosomes to this factor of co-translational protein folding.

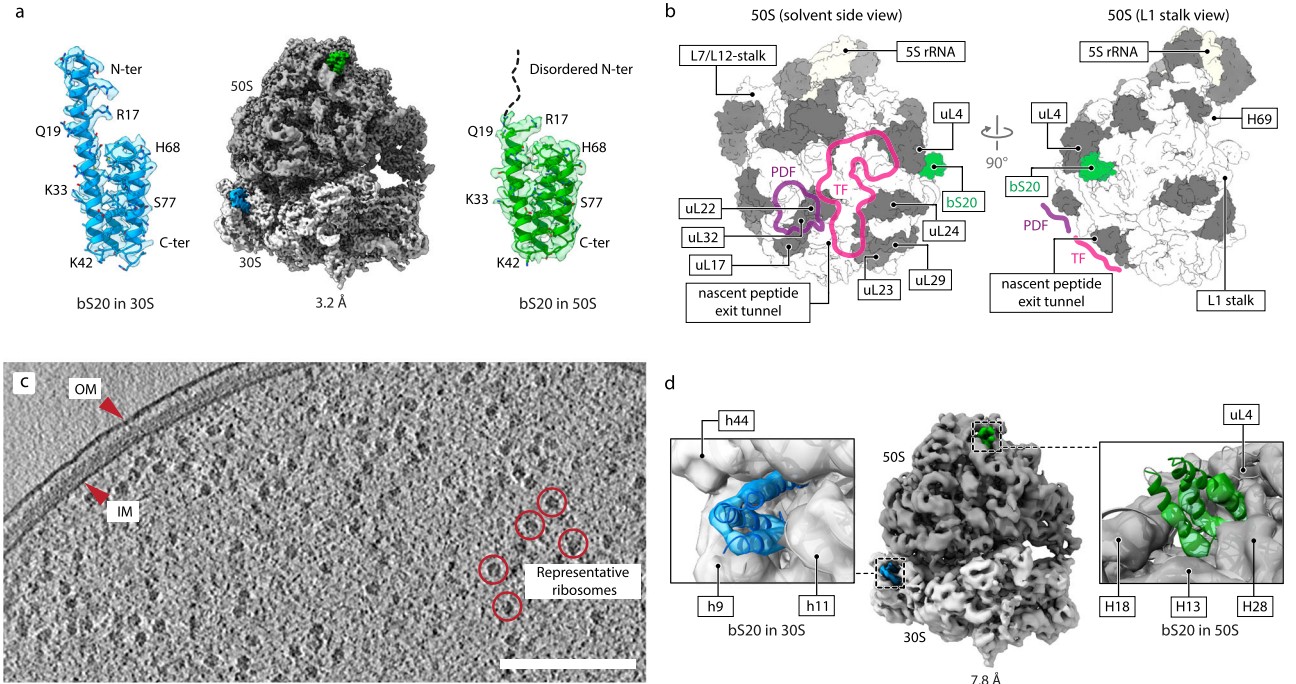

**Fig. 1 | bS20 is a structural component of the large ribosome subunit. a** Cryo-EM structure of 70S ribosomes isolated from cold-shocked *P. urativorans* demonstrates that ribosomal protein bS20 can bind to both small and large ribosomal subunits. Close-up views show the maps and molecular models of bS20 in the 30S ribosomal subunit (left) and the 50S ribosomal subunit (right). **b** Location of the novel bS20-binding site on the large subunit. bS20 is located on the solvent side of the large subunit, far away from the ribosome functional centres, such as the nascent peptide exit tunnel and the binding sites for factors of co-translational protein processing and folding (trigger factor (TF) or protein deformylase (PDF)). **c** Slice through a representative tomogram depicting an exponentially growing *P. urativorans* cell. The inner (IM) and outer (OM) membranes and representative ribosomes are indicated. Scale bar: 200 nm. In total, 28 tomograms were analyzed in this study and showed similar results. **d** Cryo-ET reconstruction of 70S ribosomes within exponentially growing bacterial cells confirms that this previously unknown bS20-binding site on the large subunit is utilised in vivo.

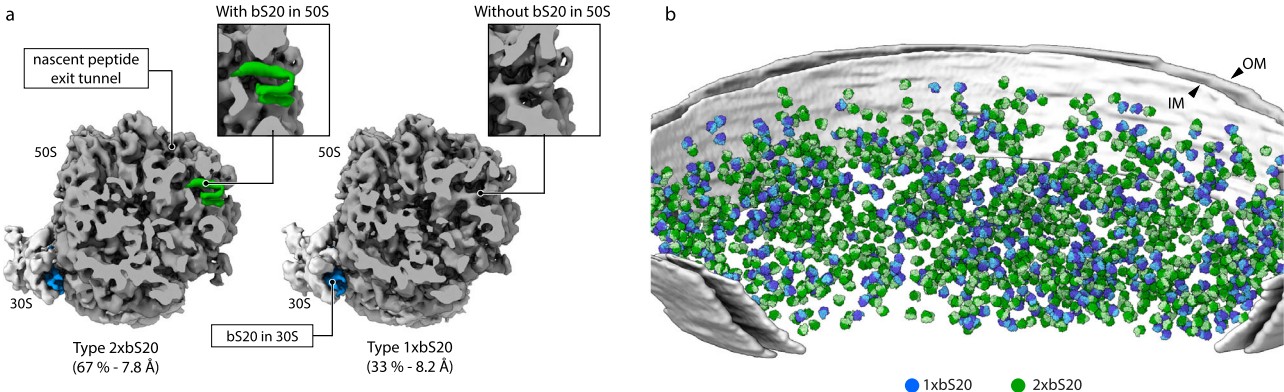

**Fig. 2 | bS20 creates two distinct ribosome types in cells. a** Cryo-ET maps corresponding to focused-classified 70S ribosomes with (left) and without bS20 (right) in the large subunit. **b** Cryo-ET imaging of *P. urativorans* cells from exponentially growing cultures illustrates the spatial distribution of 1xbS20 (blue) and 2xbS20 ribosomes (green) in bacterial cells. The inner (IM) and outer (OM) cell membranes are indicated.

This analysis revealed that both 1xbS20 and 2xbS20 ribosomes had identical ratios between the bound (~88%) and free state (~12%), highlighting that the second copy of bS20 did not influence ribosome association with Trigger Factor (Fig. 3b; Supplementary Figs. 6–8).

Next, we tested whether structurally heterogeneous ribosomes can cooperate by translating the same mRNA molecules. During protein synthesis, ribosomes associate into large supramolecular assemblies, termed polysomes, in which multiple ribosomes simultaneously translate a single mRNA molecule to produce proteins[52]. In recent cryo-electron tomography studies of ribosomes, the native structure of polysomes was inferred based on the analyses of relative orientations of neighbouring ribosomes (Fig. 3c)[53,54]. Following a similar approach,

we implemented our own polysome-tracking algorithm to locate polysomes in bacterial cells and quantify the distribution of 1xbS20 and 2xbS20 ribosomes in polysomes in situ (Methods). This analysis revealed that both types of ribosomes co-formed polysomes, with average ratio of 2xbS20 ribosomes of 67% (+/- 22% standard deviation), which mirrored the occupancy of bS20 in the large ribosomal subunit of all cellular ribosomes (Fig. 3d).

In addition, we also measured the distances separating mRNA exit and entry sites of neighbouring polysomal ribosomes to indirectly estimate the apparent relative elongation rates of 1xbS20 and 2xbS20 ribosomes. We assessed pairwise distance distributions between the mRNA entry and mRNA exit of all neighbouring ribosomes to trace

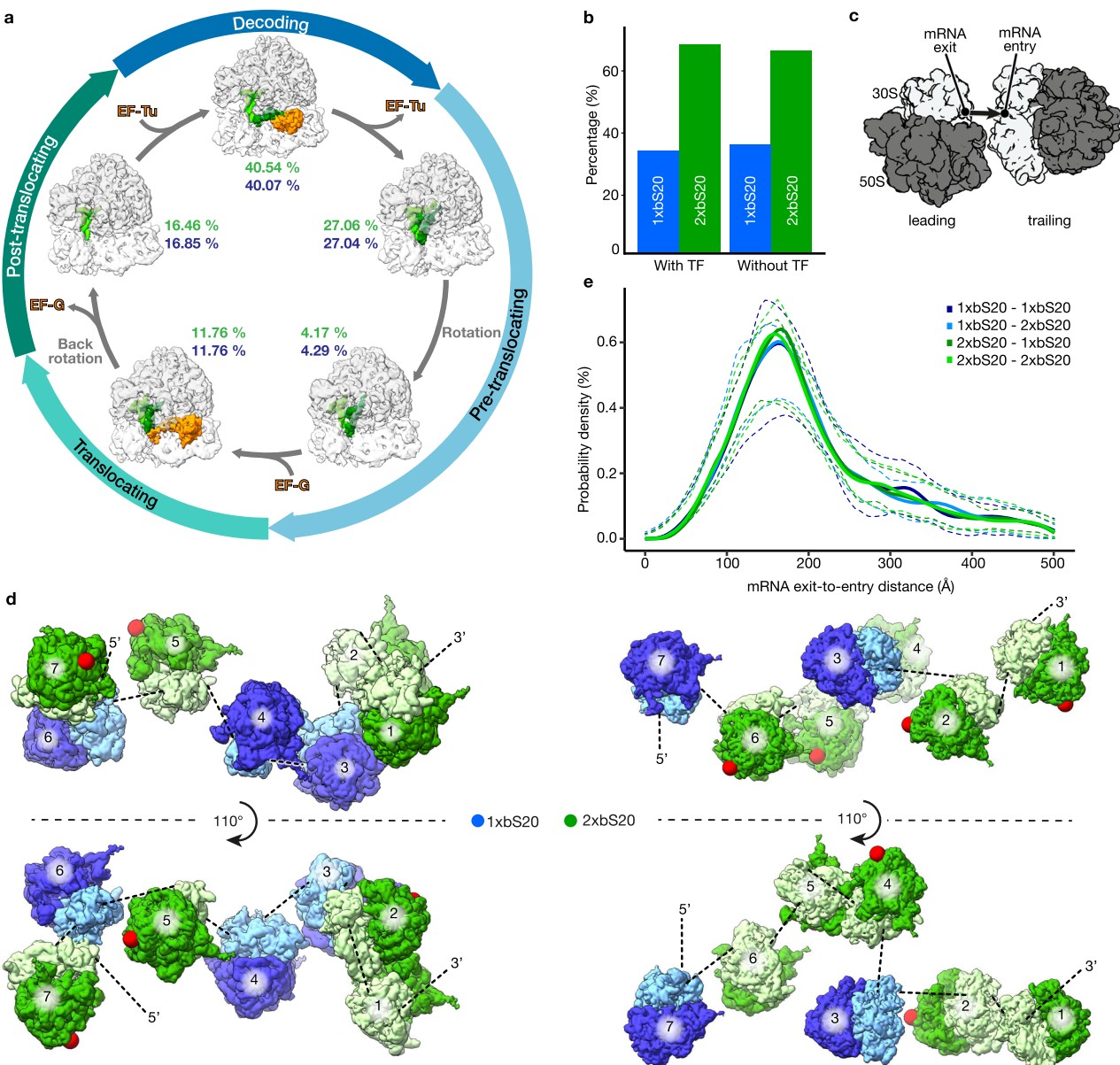

**Fig. 3 | Structural variations caused by bS20 are functionally neutral. a** In situ classification of ribosomes according to their functional states shows that 1xbS20 (blue) and 2xbS20 (green) ribosomes have the same distribution along the elongation cycle. **b** In situ classification of ribosomes according to their binding to the Trigger Factor (TF) shows that 1xbS20 (blue) and 2xbS20 (green) ribosomes have the same propensity to bind the Trigger Factor in situ. Ribosomes from 28 tomograms were analysed. **c** Cartoon representation of two adjacent ribosomes with the mRNA exit-to-entry trajectory underlying polysome tracing highlighted (see Methods for detailed description). **d** 3D visualization of two representative polysomes from cellular tomograms illustrates that 1xbS20 (blue) and 2xbS20 (green) ribosomes co-occur on the same mRNA molecules. Red spheres indicate the position of bS20 on the large subunit of 2xbS20 ribosomes. **e** Distribution of distances between the mRNA exit of any ribosome to the mRNA entry of its closest polysomal neighbour in *P. urativorans* cells. The mRNA exit-to-entry distance was plotted for each possible neighbour combination of 1xbS20 and 2xbS20 ribosomes. Dashed lines indicate the 5% and 95% confidence intervals derived from cell-to-cell variation (measured using 28 individual cells). **a–e** were obtained using exponentially growing *P. urativorans* cells. Source data are provided in a source data file.

polysomes in 28 individual cells of *P. urativorans* (Fig. 3e). We reasoned that if ribosomes bearing one or two copies of bS20 move at significantly different rates along the mRNA, we would expect to see collisions on polysomes containing both types of ribosomes. For example, if 2xbS20 ribosomes are translating slower, we would expect to find 1xbS20 ribosomes accumulating behind 2xbS20 ribosomes - thereby changing the distributions for pairwise inter-ribosome distances on all polysomes in the cell. Indeed, ribosome queuing (disome and trisome formation) has been demonstrated on ribosome quality control (RQC) substrates and at frameshift sites in bacteria and eukaryotes[55–59]. However, we did not observe these collisions, or any statistically significant differences between inter-ribosome distances (Fig. 3e). Instead, we found that the

presence of bS20 in the large ribosomal subunit has no measurable impact on inter-ribosome distances, suggesting that 1xbS20 and 2xbS20 have identical elongation properties in protein synthesis. Overall, these analyses provide direct evidence that heterogeneous ribosomes can cooperate during protein synthesis on the same molecule of mRNA, and that the presence of bS20 in the large subunit represents a functionally neutral intrinsic variation to ribosome structure.

## The bS20-binding site in the large subunit is highly conserved in proteobacteria
Finally, we asked what governs the preferential binding of bS20 to the small subunit and whether its binding to the large subunit is conserved

in other bacteria. To answer this, we first compared the bS20-binding sites in the large and small subunits by measuring the ribosome-buried area for both bS20 copies. We found that bS20 makes nearly twice as many contacts in the small subunit as in the large subunit (2510 Å$^2$ compared to 1325 Å$^2$) (Fig. 4a–c). Our mapping of the ribosome-binding residues of bS20 showed that 43 residues of bS20 form direct molecular interactions with the small subunit. However, when bS20 binds to the large subunit, only 15 of these 43 residues bind the ribosome, with the remaining 28 no longer involved in ribosome recognition (Fig. 4). This difference suggests a weaker affinity of bS20 to the large subunit, thus explaining its preferential binding to the small subunit.

Furthermore, we found that the interactions of bS20 with the small and large subunits have different physical natures. In the small subunit, bS20 interacts exclusively with rRNA, but in the large subunit, bS20 interacts largely with another ribosomal protein, uL4 (628 of 1325 Å$^2$ of the bS20-binding interface). These interactions are electrostatic and mediated by the basic residues in bS20 and the acidic residues in uL4 (Fig. 4d, e). Our evolutionary analysis shows that the bS20 sequence is highly conserved in bacteria, but the bS20-binding residues in protein uL4 are conserved only in proteobacteria (Fig. 4f, g). Thus, bS20 binding to the large subunit is likely a hallmark feature of proteobacterial ribosomes, distinguishing them from ribosomes of other bacteria.

## Discussion

### Heterogeneous ribosomes do not segregate but cooperate during protein synthesis

In this study, we were able to directly observe individual structurally heterogeneous ribosomes during protein synthesis within single bacterial cells. We have determined their molecular composition, functional states and cooperative behaviour in situ. Through this comprehensive analysis, we found that heterogeneous ribosomes do not segregate into separate populations; instead, they function equivalently and cooperatively while co-occurring on the same mRNA molecules in situ. Thus, ribosome heterogeneity does not necessarily lead to functional specialization, even when it involves significant variations, such as the presence or absence of a ribosomal protein.

This finding provides counterevidence to the prevailing view on heterogeneous ribosomes. Since their discovery more than fifty years ago, the field has been dominated by the notion that if ribosomes are structurally heterogeneous it suggests that ribosomes may be functionally heterogeneous as well[5]. However, these ideas are difficult to reconcile with many key characteristics of ribosomes. For example, heterogeneous ribosomes produced by a given organism exhibit much smaller variations compared to those observed across species. In eukaryotic organisms such as *Saccharomyces cerevisiae*, two alternative isoforms are expressed for 31 ribosomal proteins, and these isoforms are considered potentially specialized[60]. However, many of them differ from each other by only a single amino acid substitution; for instance, isoforms of protein uL4 are 99.7% identical. In contrast, these isoforms contain 88 substitutions compared to uL4 from another yeast, *Candida albicans* (75.4% sequence identity). Nevertheless, uL4 is regarded as functionally conserved, not specialized, between these two yeast species. At the same time, synthetic biologists have shown that natural ribosomes require substantial unnatural modifications to achieve the functional autonomy of heterogeneous ribosomes in terms of spatial segregation and preferential binding to mRNAs[61–64]. These modifications include mutations of the anti-Shine-Dalgarno sequence and the chemical tethering of ribosomal subunits.

Consequently, the idea that naturally occurring heterogeneous ribosomes can act in a specialized manner within a single cell lacks direct examination, due to a previous gap in technologies for observing individual heterogeneous ribosomes in situ. Here, we provide this direct examination and demonstrate that, in contrast to specialization,

ribosomes can withstand even significant structural variations, such as the presence of an additional protein, and retain their core activities in situ. Thus, cellular ribosomes can tolerate fluctuations in their molecular structure without losing their essential and cooperative activities required for ribosomal protein synthesis.

### Ribosome structures indeed have a variable stoichiometry in single cells

Our discovery of bS20 variations reveals a mechanism for the variable stoichiometry of ribosome structures. Currently, two contradicting lines of evidence exist regarding the molecular composition of ribosomes. On one hand, structural data suggest that ribosomes maintain a strictly defined stoichiometry, with each ribosomal protein present in a single copy per ribosome (with the exception of proteins L7/L12 and P1/P2, which are found in four or six copies)[28]. Conversely, biochemical and proteomic studies indicate that certain ribosomal proteins can exceed a single copy per ribosome[2,3,65,66]. Whilst older methods lacked quantitative precision, the technologies for protein analysis have developed significantly during the last few decades[67–69]. Furthermore, advancements in quantitative proteomics have revealed that similar variations may occur in eukaryotic ribosomes[17,18]. Nevertheless, the origins and mechanisms of these variations remain unknown, and it is still debated whether they are even structurally possible. Consequently, the following key questions remain unanswered: How do ribosomes achieve stoichiometric heterogeneity to contain more than one copy of a certain protein? And if this occurs, do these stoichiometric variations affect the ribosome activity?

In this study, we resolve this mechanism and the controversy for protein bS20. In contrast to previous studies suggesting that occasional observations of bS20 in the large subunit are a misannotation or a ribosome isolation artifact[44–47], our work shows that bS20 is a genuine component of the large ribosomal subunit in *P. urativorans*, and likely also in other proteobacteria. More generally, our work illustrates one possible way in which variable stoichiometry of a ribosomal protein can be achieved structurally. We suggest that the underlying cause for this heterogeneity may be related to excess ribosomal proteins in bacterial cells. Previously, the expression of ribosomal proteins was shown to be tightly regulated to ensure their production in a 1:1 ratio relative to each other[70]. This regulation helps prevent the accumulation of free ribosomal proteins, which can cause stress due to their non-specific interactions with negatively charged molecules within the cell. However, achieving a 1:1 production ratio is a non-trivial challenge because ribosomal proteins are encoded by 18 different operons[71]. Our finding of bS20 in the large subunit shows that bacterial cells may deposit excess bS20 directly into ribosomal structures. Therefore, it will be intriguing to test in the future if this bS20 binding allows bacteria to sequester free fractions of ribosomal proteins to prevent deleterious effects on cell function.

## Methods

### Cryo-EM analysis of *P. urativorans* ribosomes

To analyse bS20 association with the ribosome, 10 µL aliquots of crude ribosome samples were isolated from ice-treated *P. urativorans* (ATCC 15174). Actively growing *P. urativorans* cells (OD$_{600}$ ~ 0.3) were chilled on ice for 30 min, centrifuged at 5000 g for 10 min at 4 °C, and 1 g of cells were resuspended in 1 ml of buffer A (50 mM Tris-HCl pH 7.5, 20 mM magnesium acetate, 50 mM KCl). The suspension was transferred to tubes with 0.1 ml zirconium beads (Sigma-Aldrich BeadBug) and disrupted using a bead beater (Thermo FastPrep FP120 Cell Disrupter) at 6.5 m/s for 30 s. After centrifugation at 16,000 g for 5 min at 4 °C, the supernatant was collected and centrifuged again to remove debris. To isolate ribosomes, PEG 20,000 (A17925-0B, Thermo Fisher Scientific) was added to the lysate (0.5% w/v), and insoluble aggregates were removed by centrifugation. PEG concentration was then increased to 12.5% (w/v) to precipitate ribosomes using centrifugation

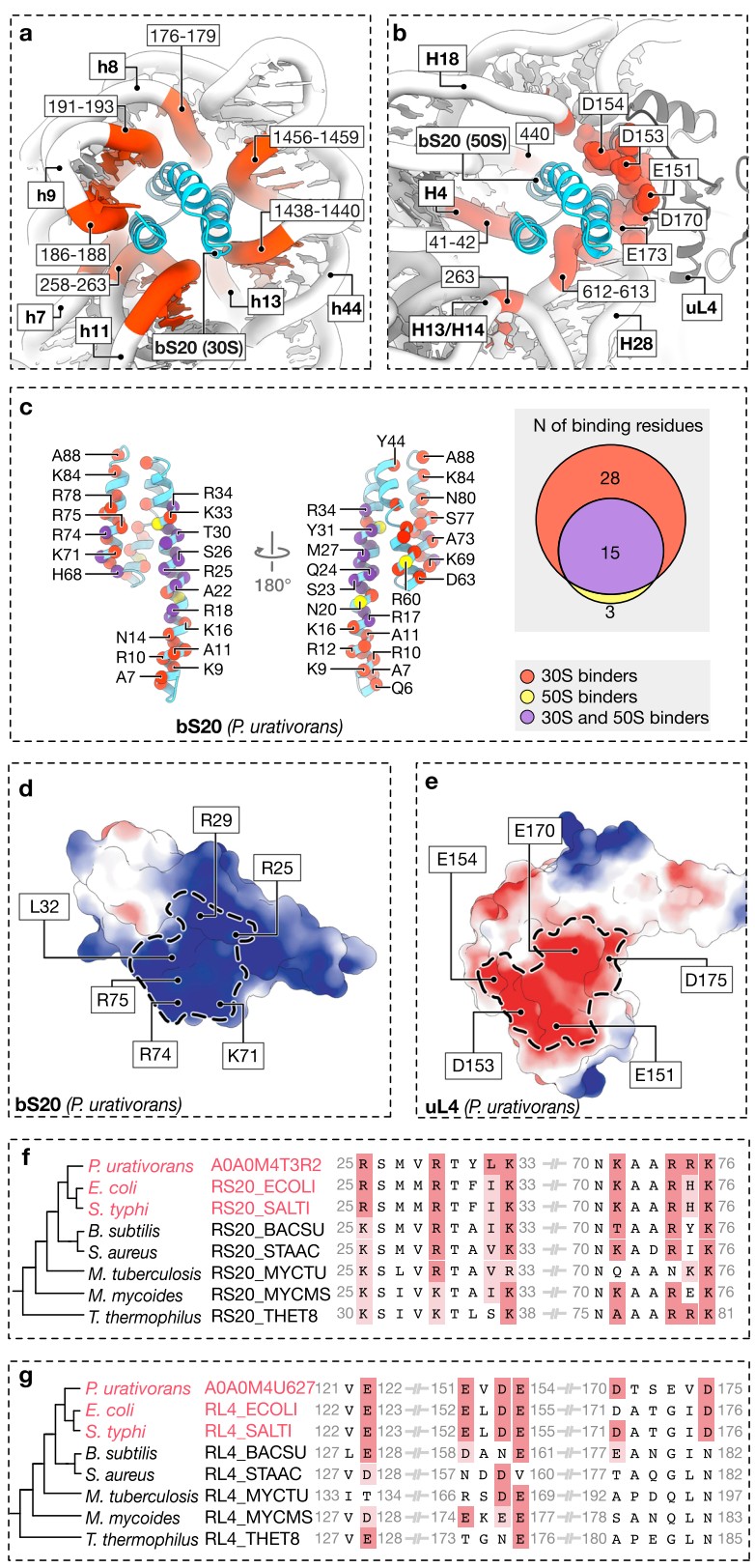

**Fig. 4 | The bS20-binding site in the large subunit is conserved in proteo-bacteria. a, b** Close-up views of the molecular interactions of bS20 in the small subunit (**a**) and the large subunit (**b**), as observed in the high-resolution cryo-EM structure of the *P. urativorans* ribosome (Fig. 1a). **c** Details of bS20 amino acid residues that mediate interaction with *P. urativorans* ribosomes. Residues are color-coded based on whether they are involved in binding to both subunits "30S and 50S binders" (purple), only the small subunit "30S binders" (red), or only the large subunit "50S binders" (yellow). **d, e** Electrostatic surface potential of bS20 (**d**) and uL4 (**e**) based on structural models of the proteins in the ribosome. The uL4/bS20 contact area is shown as dashed outlines. **f, g** Multiple protein sequence alignment illustrates sequence conservation of bS20/uL4-interface residues. uL4-binding residues of bS20 are highly conserved across bacterial species (as shown in **f**), but the bS20-binding residues of uL4 are conserved only in proteobacteria (as shown in **g**).

at 16,000 g for 5 min at 4 °C. The ribosome pellet was dissolved in 50 µl of buffer A, and the solution was purified using PD Spin Trap G-25 columns (GE28-9180-04, GE Healthcare), yielding a ribosome concentration of ~500 nM. Aliquots were stored at −20 °C for cryo-EM analysis.

The ribosome sample was thawed on ice and an aliquot of 2 µL was applied onto glow discharged (20 mA, 90 s, 0.26 mBar, PELCO easiGlow) Quantifoil grids (R1.2/1.3, 200 mesh, copper). Grids were blotted for 1 s (using blotting force -5) at 100% humidity, 4 °C and vitrified using liquid nitrogen-cooled ethane in a Vitrobot Mark IV (Thermo Fisher Scientific). This grid was then used for data collection with a 200 kV Glacios electron cryo-microscope (Thermo Fisher Scientific) with Falcon 4 detector located at the York Structural Biology Laboratory, University of York, UK. For each movie a total dose of 50 e-/Å² was applied to the grid across 7 s. A nominal magnification of 150,000 x was applied, resulting in a final object sampling of 0.934 Å pixel size. 3706 micrograph movies were recorded in aberration free image shift (AFIS) mode using defocus targets of -1.25, -1.0, -0.75, -0.5 µm.

Cryo-EM data for the *P. urativorans* dataset was processed as summarized in Supplementary Fig. 1 and Supplementary Table 2. In brief, using RELION 3.1[72] a total of 303,839 particles were picked from 3707 micrographs using the Laplacian of Gaussian picker (160 – 300 Å; 0.9 standard deviation threshold). Particles were then extracted using a 450 px box with no downscaling. After two rounds of 2D classification were carried out to clean the dataset, 23,366 "good" particles were selected for 3D refinement. This resulted in a map at 4.5 Å resolution. Subsequently, CTF refinement (per particle), aberration correction and postprocessing was carried out to generate a map at 3.2 Å resolution. This map clearly showed heterogeneity around the bS20-binding site in the large ribosomal subunit due to partial occupancy of bS20. To resolve this, using RELION 5.0[73] angular assignments from the previous refinement job were used to carry out masked classification without alignment focusing on the bS20-binding site in the large ribosomal subunit. This separated particles into four groups corresponding to differential factors of occupancy: classes 1 and 2 -small subsets of bad particles; class 2-large subunit bS20(occupied); class 3-large subunit bS20(vacant). The final particle subsets of "large subunit bS20(occupied)" and "large subunit bS20(vacant)" were selected for masked 3D refinement and postprocessing resulting in maps with a 3.2 and 6.7 Å resolution respectively. Finally, sharpened maps weighted by estimated local resolution were calculated. All reported estimates of resolution are based on gold standard Fourier shell correlation (FSC) at 0.143, and the calculated FSC is derived from comparisons between reconstructions from two independently refined half-sets (Supplementary Fig. 2).

## Model building, refinement, deposition

The atomic model of *P. urativorans* ribosomes was retrieved from the Protein Data Bank (PDB ID 8RD8). The model was then modified by deleting the ribosome ligands, such as EF-Tu, Balon, and RaiA, due to their partial occupancy in our cryo-EM map. The density corresponding to bS20/L26 in the large ribosomal subunit was initially modelled as a poly-alanine chain in the single-particle cryo-EM reconstruction of the *P. urativorans* ribosome (at 3.2 Å resolution) to determine the backbone structure. This poly-alanine backbone model was then used to find proteins with similar fold in the Alpha Fold bank of predicted structures using the FoldSeek tool for tracking structural similarities of macromolecules[50]. To create the final model of bS20 in the large subunit, the starting atomic model of bS20 was extracted from the structure of *P. urativorans* small ribosomal subunit (PDB ID 8RD8) and edited using Coot v0.8.9.2[74]. The obtained structure of *P. urativorans* ribosome bearing bS20 in both the large and small ribosomal subunits was refined using Phenix 1.20.1[75] and deposited in the Protein Data Bank (PDB ID 9HC4).

## Automated model building

To provide additional validation of our annotation of the unknown protein as bS20, we used ModelAngelo v1.0.13[51] to automatically build a model into the unidentified density in the large ribosomal subunit. The region of interest in the postprocessed cryo-EM map was masked, and, to avoid bias, ModelAngelo was run without any input sequences. ModelAngelo produced a partial model of bS20, corresponding to three helices. Sequence alignment using the EMBL-EBI sequence analysis tool[76] against the bS20 sequence as a reference indicated that these helices correspond to residues 17-35 in the α1 helix residues 47-49 and 57- 66 in the α2 helix and residues 72-74 and 81-83 in the α3 helix. ChimeraX v1.8[77] was then used to superimpose the partial ModelAngelo model with the structure of bS20 in the small ribosomal subunit (PDB ID 8RD8). RMSD values were further used to assess the structural similarity between both models, as reported in Supplementary Fig. 4.

## Sample preparation for cryo-ET

*P. urativorans* cells were cultured in nutrient broth #3 (Beef Extract 3.0 g/L Peptone 5.0 g/L) at 19 °C with shaking at 130 rpm until plunge freezing once the culture reached an $OD_{600} = 0.58$. For plunge freezing, EM support grids (Cu R2/1, mesh 200; Quantifoil) were glow discharged using a Solarus 950 plasma cleaner (Gatan) for 30 s under oxygen atmosphere. 4 µL of cell suspension were applied to the carbon side of the grid inside the chamber of a Vitrobot Mark IV (Thermo Fisher Scientific) that was kept at 19 °C. Excess liquid was removed by blotting for 2 s using a filter paper facing the backside and a Teflon sheet facing the carbon side of the grid. Electron transparent lamellae where prepared by cryo-FIB milling using an Aquilos 2 microscope (Thermo Fisher Scientific). Grids were screened using the software Maps (Thermo Fisher Scientific) and milling positions were chosen at positions with clusters of cells in sufficiently thick ice. Samples were coated with an organometallic platinum layer using a gas injection system for 40 s followed by sputter coating with platinum at 30 mA and 10 Pa for 10 s. Milling was performed using the software AutoTEM (Thermo Fisher Scientific) to thin the positions in four steps followed by two steps of polishing to a target thickness of 140 nm.

## Cryo-ET data collection

A total of 39 tilt series were acquired in two independent sessions from 14 lamellae on 2 grids. Data was collected on a Titan Krios (Thermo Fisher Scientific) operated at 300 kV, equipped with a Gatan K3 direct electron detector and a Quantum Gatan Imaging Filter (Gatan) with an energy slid width of 20 eV. Tilt Series were acquired using Tomo5 (Thermo Fisher Scientific) at 33,000x magnification corresponding to a pixel size of 2.589 Å. The acquisition was started from the lamella pretilt of 9°–12° and a dose symmetric scheme with 3° increments from -50° to 60° was applied resulting in 38 projections per tilt series with a total dose of 136 e-/Å² and a dose rate of 26 e-/px/s. The nominal defocus was varied between -3 to -6 µm.

## Subtomogram analysis of *P. urativorans* ribosomes

Data were processed as summarized in Supplementary Fig. 6. Individual micrographs were motion corrected using MotionCor2[78] with 5 × 5 patches. Subsequently, CTF parameters were estimated using Warp[79] and image stacks were created for subsequent tilt series alignment using IMOD version 4.11.19[80]. The alignment parameters were used to reconstruct binned tomograms with a pixel size of 20.712 Å in Warp. For particle localization a template was generated using fast rotational matching alignment implemented in PyTom[81] from approximately 500 manually selected ribosomes. Template matching was performed as implemented in PyTom. The top 2000 cross-correlation peaks were extracted per tomogram, and obvious contamination was removed manually in UCSF Chimera. Subtomograms were reconstructed in Warp (2.589 Å/px) and classified in RELION 3.1[72] to remove non-

ribosomal particles. This yielded 40,853 ribosomal particles which were used to correct the tilt series for local deformation in M version 1.0.9[82] prior to reconstructing corrected subtomograms (2.589 Å/px). These subtomograms were subjected to focused classification on the 30S subunit using a smooth shape mask to sort for rotational states. Rotated (6245 particles) and unrotated (31,444 particles) 70S ribosomes were subclassified separately by focusing on the factor and tRNA binding sites to assign specific translational states. For the translocating state, two EF-G states were observed (pre- and post-GTP-hydrolysis) and pooled for analysis. Low quality classes were excluded at each classification step, leaving 37,352 high quality particles.

To identify the number of ribosomes with a second copy of bS20 bound to the 50S subunit, 3D classification was focused on the 50S subunit binding site of bS20 with a small spherical mask and providing a reference without bS20 on the 50S subunit. To assess ribosome association with Trigger Factor, a smooth shape mask was generated covering the Trigger factor binding site in vicinity of the ribosomal tunnel exit. Focused 3D classification using this mask yielded one class lacking Trigger Factor and several classes with Trigger Factor in different conformations, which were grouped for analysis. All final classes were subjected to multi-species refinement runs in M followed by post-processing in RELION 3.1 (Supplementary Table 3). Custom scripts used for cryo-ET data analysis can be found in Supplementary Data 1.

### Polysome tracing and analysis

Polysome tracing using Python scripts followed a similar approach as published earlier[53]. In cryo-ET, individual mRNA molecules cannot be resolved. It is therefore a standard approach in cryo-ET[53,54] to computationally infer whether adjacent ribosomes are part of the same polysome or come into random proximity by analysing their relative orientations. If they are on different mRNA molecules, they will adopt rotational poses that are unconstrained (i.e. all relative orientations are equally likely). If they are on the same mRNA molecule, the mRNA exit site of the leading ribosome will be more aligned towards the mRNA entry site of the trailing ribosome.

All particles to be analysed were assigned a sequential unique identifier in the input STAR file. Two user-defined markers corresponding to the ribosomal mRNA entry and mRNA exit sites were defined in the coordinate system of the subtomogram average and mapped back into the coordinate system of the tomogram based on positions and orientations of particles as determined during template matching and subsequent 3D auto-refinement. For each mRNA exit site, the closest mRNA entry site among neighbouring particles was identified and registered ('ConStarLation_Particles.py'), if the distance between the two points was below the cut-off distance (polysome tracing: 25 nm; elongation rate analysis: 50 nm) (Supplementary Data 1). If multiple particles shared the same neighbour, only the shortest link was considered. If linked particles formed a closed circle, the longest link was removed. To assign particles to polysomal chains, the first unit of each traced chain was identified as a particle that has an upstream but no downstream neighbour. From the first unit onwards, particles were chained until no upstream neighbour was detected anymore ('StarChainer.py'). Each traced polysome chain received an identifier and the polysome length and sequence of particles within the polysome were registered in the output particle STAR file for further analysis. Based on the sequence of particles within polysomes and the class assignment to ribosomes with one or two copies of bS20, we computed the fraction of bS20 containing ribosomes per polysome. Traced polysomes were visualized with ArtiaX[83] using a reconstruction of the 70S *P. urativorans* ribosome generated in M.

### Reporting summary

Further information on research design is available in the Nature Portfolio Reporting Summary linked to this article.

## Data availability

The atomic coordinate file for the structure of *P. urativorans* ribosomes with two copies of bS20 generated in this study has been deposited in the Protein Data Bank (PDB) with the accession code 9HC4. The associated cryo-EM maps generated in this study were deposited to the Electron Microscopy Data Bank (EMDB) with the accession codes EMD-52036 (cryo-EM reconstruction of 2xbS20 ribosomes), EMD-52351 (subtomogram average of all ribosomes), EMD-52352 (subtomogram average of 1xbS20 ribosomes) and EMD-52354 (subtomogram average of 2xbS20 ribosomes). A representative tomogram has been deposited to the EMDB with the accession code EMD-52842. Source data are provided with this paper.

## Code availability

Custom scripts used in this study are available as Supplementary Data 1.

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

## Acknowledgements

We would like to thank Bert van den Berg and Wyatt Yue (Newcastle University, UK) and Yuri Polikanov (UIC, USA) for their critical and encouraging comments on this manuscript. The authors acknowledge support by the state of Baden-Württemberg through bwHPC and the German Research Foundation (DFG) through grant INST 35/1597-1 FUGG. The authors gratefully acknowledge the data storage service SDS@hd supported by the Ministry of Science, Research and the Arts Baden-Württemberg (MWK) and the German Research Foundation (DFG) through grant INST 35/1503-1 FUGG. We would like to acknowledge access to the infrastructure and support provided by the Cryo-EM Network at the Heidelberg University (HDcryoNet), which is funded and supported by the German Research Foundation (DFG), the Federal Ministry of Education and Research (BMBF) and the Ministry of Science Baden-Württemberg, among others, within the framework of the Excellence Strategy of the Federal and State Governments of Germany. This project was undertaken on the Viking Cluster, which is high-performance compute facility provided by the University of York. We are grateful for computational support from the University of York High Performance Computing service, Viking, and the Research Computing team, and support from the Newcastle University Structural Biology Laboratory. We also acknowledge the York cryo-EM facility supported by Wellcome Trust (206161/Z/17/Z). This work was supported by the Newcastle University Overseas Research Scholarship and the Scientific Exchange Grant by the European Molecular Biology Organization (both to K. H.-B.), the BBSRC Doctoral Training Programme (BB/T008695/1 to L.I.C.), the European Research Council (ERC-StG 'RiboStress' to S.P.), the Aventis Foundation (to S.P.), the 'Chica and Heinz Schaller' Foundation (to S.P.), a Wellcome Trust and Royal Society Sir Henry Dale Fellowship (221818/Z/20/Z to C.H.H.), and the Royal Society International Partnership Grant (IES\R2\222173 to S.P. and S.V.M.).

## Author contributions

K.H.-B., S.K., S.P., C.H.H., and S.V.M. designed the study. K.H.-B. collected and processed the cryo-EM data. S.K. collected the cryo-ET data and analyzed it with assistance from S.F.. L.I.C., S.B., and M.H. assisted with bS20 identification. A.B. assisted with cryo-EM data storage, transfer, and processing. All authors contributed to the writing of the manuscript.

## Competing interests

The authors declare no competing interests.
