## [Transparent Peer Review file · Nature Communications]

Structurally heterogeneous ribosomes cooperate in protein synthesis in bacterial cells

Corresponding Author: Dr Sergey Melnikov

Version 0:

Reviewer comments:

Reviewer #1

(Remarks to the Author)

The manuscript entitled “Structurally heterogeneous ribosomes cooperate in protein synthesis in bacterial cells” ribosome structure of cold-shocked 70S ribosomes and in the cells of bacterium *Psychrobacter urativorans* was analysed by cryo-EM and tomography. Isolated ribosomes as well as ribosomes in situ were found to be heterogeneous in respect of r-protein composition. Large fraction of 70S ribosomes contain two copies of protein bS20 (formerly also known as S20/L26). One copy of the protein bS20 is on the small ribosome subunit in the previously known binding site and another copy in the large ribosome subunit. S20 binding site on the LSU was found to be highly conserved among proteobacteria. Unlike of SSU, its binding site on the LSU consist of mostly protein (uL4). The authors were able to demonstrate that both types of ribosomes (containing one or two copies of bS20) are present on the same mRNA molecule and are equally distributed on the mRNA. The experiments are performed in a careful way and interpreted correctly. The manuscript is well written and easy to read. References are correct. Articles on the ribosomal protein stoichiometry in ribosomes and its subunits are mostly original papers published over 30 years. It is important to mention that the methods of protein analysis have advanced significantly during last decades while the old methods were not very precise. Few papers using qMS (e.g. PMID: 23228329) could be added.

Criticism:

I. 50-53: “...contrary to the prevailing hypothesis, ribosome heterogeneity does not necessarily lead to functional specialization, even when it involves significant variations, such as the presence or absence of a ribosomal protein.” Functional heterogeneity of bacterial ribosomes is based on the rRNA sequence variations (mostly related to the 3' end region of SSU rRNA) according to the published papers. Only a few publications (eg PMID: 19187763) mention ribosomal proteins as a source of functional heterogeneity (see reviews PMID: 34312084 and PMID: 30733326). Therefore, it is not a “prevailing hypothesis” but rather a rare case. Note that functional significance of of MazF directed cleavage of 16S rRNA was questioned (PMID: 29861158).

Introduction: Can you add few words about the organism *Psychrobacter urativorans*?

I. 200 It could be mentioned that over 70% of the ribosomes isolated from cold-shocked bacteria contains two copies of S20. That is the same fraction as the living cells.

I. 249-251 – “In addition, we also measured the distances separating mRNA exit and entry sites of neighbouring polysomal ribosomes to indirectly estimate the relative elongation rates of 1xbS20 and 2xbS20 ribosomes.” – For readers it would be extremely helpful to bring out reference to previous works where distance between polysomes has been used to estimate elongation rates of ribosomes. Perhaps it is your own model/theory, in that case please provide more information to elucidate how these two parameters: distance between ribosomes in polysomes and elongation rate of ribosomes are connected.

I. 358-361 – “Particularly, we demonstrate that ribosomes do not have a strictly defined stoichiometry, where each protein is present in a single copy. Instead, ribosomes can have a variable protein stoichiometry, and we resolve a mechanism of this previously elusive phenomenon.” – A rather ambitious statement, considering that your work and analysis revolves around one protein, bS20, and yet the sentence generalizes that ribosome do not have strict defined stoichiometry where each protein is present in single copy. If you wish to make that kind of statement, then please provide more context! Perhaps references to the previous works with similar results for S20 and other rp's can be mentioned.

I. 286 “...contrary to the prevailing hypothesis, we showed that ribosome heterogeneity does not necessarily lead to functional specialization, even when it involves significant variations, such as the presence or absence of a ribosomal protein.” see comment to I. 50.

I.289-300. The references on the ribosome heterogeneity due to the ribosomal proteins concern exclusively Eukaryotic

ribosomes. This should be mentioned as species names in Latin are not clear to all readers. And it is an important point. l. 322-324. "For example, studies of *E. coli* under various growth conditions have shown that ribosomal proteins bL32, bL33, bL34, bS6, bS21 and bS20 can occasionally occur in at least 1.2 copies per ribosome." In the references added two-dimensional gel electrophoresis was used, a method that is not exactly quantitative. 20% variation does not definitely reflect the real stoichiometry. Please look at more recent papers.

l. 358-360. "Particularly, we demonstrate that ribosomes do not have a strictly defined stoichiometry, where each protein is present in a single copy." This has been observed repeatedly, references would be nice.

Finally, in multiple places examples: line 6, 289, and 291. The words heterogenous and heterogeneous is used numerous times with apparently same meaning. While the words are similar in their meaning, they are not the same. A quick search revealed these differences in their meanings in the field of medicine and radiology:

- o Heterogeneous refers to a structure with dissimilar components or elements, appearing irregular or variegated.
- o Heterogenous refers to a structure having a foreign origin.

Reviewer #2

(Remarks to the Author)

Reviewer #3

(Remarks to the Author)

The authors present compelling structural evidence demonstrating that in *Psychrobacter urativorans*, compositional ribosome heterogeneity associated with the ribosomal protein bS20 does not result in functional specificity among distinct ribosomal pools. Using high-resolution cryo-EM, they identified two ribosomal populations, each containing either one or two copies of bS20. This finding was corroborated through in situ cryo-electron tomography (cryo-ET) subtomogram averaging, eliminating the possibility that the observed structural variability arose from the purification process. Furthermore, cryo-ET analysis enabled precise estimation of the localization and orientation of each ribosomal species within the cellular environment. Importantly, no significant differences were observed in translational states, subcellular localization, or polysome engagement between these ribosomal populations.

The study is well-executed, with clear and logical presentation of the results. I consider this work a robust contribution to the understanding of structural ribosomal heterogeneity and its lack of direct functional specificity. The experimental design and conclusions are sound, and I recommend the manuscript for publication after minor revisions.

Minor Points for Consideration:

1. In the figures and text, the ribosomal "exit tunnel" is inconsistently referred to as the "tunnel exit." I recommend standardizing this terminology.
2. The captions for Figures 1 and 2 include concluding statements that seem more appropriate for the Results and Discussion sections. Specifically, sentences beginning with "Overall, this figure..." should be removed to avoid redundancy.
3. Supplementary Figures S1 and S2 are not referenced in the first section of the Results. Please ensure these figures are cited appropriately in the text.

Reviewer #4

(Remarks to the Author)

The manuscript of Helena-Bueno and coworkers addresses heterogeneity of translating ribosomes in prokaryote cells upon cold shock treatment. Ribosome heterogeneity within cells and its outcome in cell fate has been a well debated subject, and evidence of this physical heterogeneity has increased with state-of-the-art mass spectrometry and Cryo-EM. No longer is the ribosome simply viewed as a blindly translating machine, but has gained importance as a central player in the regulation of gene expression. The full extent of this ribosome heterogeneity and how it influences cell fate through translation is still poorly understood. Here the authors seek to investigate the heterogeneity/segregation of translating ribosomes in *Psychrobacter urativorans*, specifically concerning the small ribosomal subunit protein bs20, (which is also able to bind to large ribosomal subunit) using Cryo Electro tomography (Cryo-ET) of FIB-milled cells. Although the topic is very exciting and the technique used to address this is state of the art, overall, the manuscript appears preliminary and the data provided are mostly in-silico analysis. Authors lack a control of cells grown in normal conditions, and do not provide any biochemical data or literature complementing their Cryo-ET driven conclusions, Such as in-vitro translation with purified polysomes to support that polysomes bearing one or two copies of bs20 translate at similar rates, or ribosome profiling to assess global polysome populations in cold shock conditions. Their conclusions are solely based on in-silico analysis of their Cryo-ET lamella of flash frozen bacteria grown in cold shock inducing conditions.

Based on these remarks, I believe that the manuscript as such does not provide findings strong and new enough to be published in Nature Communications.

Major points

1. The manuscript gives the reader a strong impression that bs20 binding to the 50S is something novel, however, it has been previously shown by several independent studies performed in various bacteria. Only in the discussion sections do the authors mention and cite that bS20 was previously known to bind to the 50S subunit. Hence, the introduction and results

sections should better reflect this point.

2. Do the 14 FIB-milled lamella represent different sections of the flash frozen bacteria? Or were bacteria always FIB-milled in the same region? If bacteria were milled in the same section, how can the authors be sure that there is no ribosome segregation?
3. Related to point 2. The authors should provide a Cryo-ET analysis of bacteria grown at normal temperatures in order to demonstrate that the Ribosome population is majorly formed by 1xbs20. Alternatively, could the authors show evidences of this control experiment with purified polysomes and Cryo-EM SPA? If there are no differences between physiological temperatures and cold shock, then the notion of ribosome heterogeneity is somehow misapplied here.
4. Authors state that Polysomes bearing 1 or 2 copies of bs20, have no significant cellular segregation (assuming point 2 has been addressed properly), and their different translation rates are not altered. While the techniques used in this paper fairly support the mixed polysomes, it does not substantially support that these ribosomes translate at the same rates simply based on distances between frozen ribosomes. Other biochemical assays should be provided, or the text should be altered to fully contextualise the reader that the paper addresses this issue solely from an in-silico point of view.
5. No polysome profiling is shown (e.g. sucrose gradient) to demonstrate the global level of protein translation occurring in cells grown in cold shock or normal conditions. Is the ratio between polysomes and monosomes maintained? Do heavy/long polysomes still form?
6. Related to point 5. If possible, a Western Blot analysis of α -bs20 protein on 50S and 30S fractions originating from dissociated polysomes would also help prove the protein binding on the 50S. For instance, using puromycin in growing cells or high salt concentrations leads to 50S and 30S dissociation, which can then be purified by sucrose gradient, and fractions can be analysed by WB.
7. Related to point 5. In a flash frozen sample, how reliably can monosomes close together be differentiated from polysomes? Can the authors track mRNA in their tomograms? This could either challenge and/or partially support that the authors "protein translation" results are not biased by sudden changes in ribosome/polysome total populations (fig 3 d and S.fig 6).
8. Related to point 5 – It would be interesting to provide biochemical data e.g cell-free translation systems prepared from normal or cold shock grown bacteria, and compare translation rates
9. In figure 3 panel c, it could be interesting to highlight bs20 bound to the 50S, to understand whether it is at the interface between ribosomes and somehow helps or impede translation process.
10. Authors constructed a polyalanine based secondary structure in the bs20 (50S) map region, then searched in alpha fold seek for similar folds, yielding a list with top candidates representing bs20 protein from different bacteria. The authors further support this data with Mass Spec data analysis. Nevertheless, does the fold seek search outputs other proteins which could fit in this density? Also, given the importance of this aspect for the whole paper, a figure with more detailed information should be provided to convince the reader that the bs20 density (50S side) confidently fits bs20 side chains.
11. In figure S4a., authors compare local resolution plotted on Cryo-ET maps of ribosomes bearing 1 or 2 copies of bs20. In panel a 1xbs20 ribosomes globally point to a higher overall flexibility of solvent exposed surfaces, especially the RNA helix that in close proximity with bs20 binding site (the figure is small so it is hard to interpret correctly), as compared to 2xbs20 ribosomes. Can the authors comment on this? Would this also not affect the accurate determination of bs20 density on the 50S side, compromising the population distributions suggested by the authors?

Minor points:

Figure S1 is mentioned in the text before Figure S3;

In Figure 1 "(OM)" is duplicated;

Add resolution to relevant figures;

In Figure 3 consider adding in the subtitles the pymol script used for polysome detection; In figure 4 consider Changing "Zoomed-in" by "close-up view". I assume both models are from Cryo-ET reconstructions? If so, add this info in legend.

In Figure S4 consider circling the region where bs20 (on the 50S side) is located.

In Figure S5 , consider colouring maps for easier interpretation (e.g. 50S and 30S)

Reviewer #5

(Remarks to the Author)

Version 1:

Reviewer comments:

Reviewer #1

(Remarks to the Author)

The authors have made changes as requested. I have no more questions.

Reviewer #2

(Remarks to the Author)

Reviewer #3

(Remarks to the Author)

We thank the authors for their detailed and thorough reply to our comments. Karla Helena-Bueno and colleagues addressed our concerns and answered our questions carefully and adequately. The new version of their manuscript is now clearer, and we agree with its publication as is.

I co-reviewed this new version of the manuscript with one of the reviewers who provided the listed reports. This is part of the Nature Communications initiative to facilitate training in peer review and to provide appropriate recognition for Early Career Researchers who co-review manuscripts.

Reviewer #4

(Remarks to the Author)

We thank the reviewers for their time and consideration and provide below our point-by-point response:

Reviewer #1 (Remarks to the Author):

The manuscript entitled “Structurally heterogeneous ribosomes cooperate in protein synthesis in bacterial cells” ribosome structure of cold-shocked 70S ribosomes and in the cells of bacterium *Psychrobacter urativorans* was analysed by cryo-EM and tomography. Isolated ribosomes as well as ribosomes in situ were found to be heterogeneous in respect of r-protein composition. Large fraction of 70S ribosomes contain two copies of protein bS20 (formerly also known as S20/L26). One copy of the protein bS20 is on the small ribosome subunit in the previously known binding site and another copy in the large ribosome subunit. S20 binding site on the LSU was found to be highly conserved among proteobacteria. Unlike of SSU, its binding site on the LSU consist of mostly protein (uL4). The authors were able to demonstrate that both types of ribosomes (containing one or two copies of bS20) are present on the same mRNA molecule and are equally distributed on the mRNA.

The experiments are performed in a careful way and interpreted correctly. The manuscript is well written and easy to read. References are correct.

We thank the reviewer for these kind remarks.

Articles on the ribosomal protein stoichiometry in ribosomes and its subunits are mostly original papers published over 30 years. It is important to mention that the methods of protein analysis have advanced significantly during last decades while the old methods were not very precise. Few papers using qMS (e.g. PMID: 23228329) could be added.

We have added three additional references (Refs. 60-62), including the one suggested by the reviewer, and have now mentioned in our revised discussion (p.12):

Conversely, biochemical and proteomic studies indicate that certain ribosomal proteins can exceed a single copy per ribosome (2,3,58,59). Whilst older methods lacked quantitative precision, the technologies for protein analysis have developed significantly during the last few decades (60–62).”

Criticism:

I. 50-53: “...contrary to the prevailing hypothesis, ribosome heterogeneity does not necessarily lead to functional specialization, even when it involves significant variations, such as the presence or absence of a ribosomal protein.”

Functional heterogeneity of bacterial ribosomes is based on the rRNA sequence variations (mostly related to the 3' end region of SSU rRNA) according to the published papers. Only a few publications (eg PMID: 19187763) mention ribosomal proteins as a source of functional heterogeneity (see reviews PMID: 34312084 and PMID: 30733326). Therefore, it is not a “prevailing hypothesis” but rather a rare case.

We have addressed this by removing the “contrary to the prevailing hypothesis”:

“ribosome heterogeneity does not necessarily lead to functional specialization, even when it involves significant variations, such as the presence or absence of a ribosomal protein.”

Note that functional significance of MazF directed cleavage of 16S rRNA was questioned (PMID: 29861158).

We added the reference questioning the MazF study (Ref24).

Introduction: Can you add few words about the organism *Psychrobacter urativorans*?

We have provided this information in our Results:

*“[...] *Psychrobacter urativorans*, the cold-adapted γ -proteobacterium typically found in Arctic ornithogenic soil (48).”*

I. 200 It could be mentioned that over 70% of the ribosomes isolated from cold-shocked bacteria contains two copies of S20. That is the same fraction as the living cells.

We have addressed this by revising the following sentence in our Results:

“This revealed similar proportions of bS20 (~77%) bound to the large subunit, indicating that this binding is biochemically stable and likely independent of the growth conditions because of the similar levels of bS20 in ribosomes from cold-shocked and actively growing cells.”

I. 249-251 – “In addition, we also measured the distances separating mRNA exit and entry sites of neighbouring polysomal ribosomes to indirectly estimate the relative elongation rates of 1xbS20 and 2xbS20 ribosomes.” – For readers it would be extremely helpful to bring out reference to previous works where distance between polysomes has been used to estimate elongation rates of ribosomes. Perhaps it is your own model/theory, in that case please provide more information to elucidate how these two parameters: distance between ribosomes in polysomes and elongation rate of ribosomes are connected.

We have addressed this by expanding **Fig. 3** (panels c-e) in our manuscript and the corresponding text in our Results:

“If ribosomes bearing one or two copies of bS20 move at significantly different rates along mRNA, it follows that we would expect to see collisions on polysomes containing both types of ribosomes. For example, if 2x bS20 ribosomes were to translate slower, we would expect to find 1x bS20 ribosomes accumulating behind 2x bS20 ribosomes - thereby changing the distributions for pairwise inter-ribosome distances on all polysomes in the cell. Indeed, ribosome queuing (disome and trisome formation) has been demonstrated on RQC substrates and at frameshift sites in bacteria and eukaryotes (55-59). However, we do not observe these collisions, or any statistically significant differences between inter-ribosome distances.”

I. 358-361 – “Particularly, we demonstrate that ribosomes do not have a strictly defined stoichiometry, where each protein is present in a single copy. Instead, ribosomes can have a variable protein stoichiometry, and we resolve a mechanism of this previously elusive phenomenon.” – A rather ambitious statement, considering that your work and analysis

revolves around one protein, bS20, and yet the sentence generalizes that ribosome do not have strict defined stoichiometry where each protein is present in single copy. If you wish to make that kind of statement, then please provide more context! Perhaps references to the previous works with similar results for S20 and other rp's can be mentioned.

To address this comment, we have deleted the last section of our discussion.

I. 286 "...contrary to the prevailing hypothesis, we showed that ribosome heterogeneity does not necessarily lead to functional specialization, even when it involves significant variations, such as the presence or absence of a ribosomal protein." see comment to I. 50.

We have addressed this in our response to the comment to I.50 by deleting the "contrary to the prevailing hypothesis".

I.289-300. The references on the ribosome heterogeneity due to the ribosomal proteins concern exclusively Eukaryotic ribosomes. This should be mentioned as species names in Latin are not clear to all readers. And it is an important point.

To address this comment here and throughout the manuscript, we have reworded our text to stress that the references on the ribosome heterogeneity concern exclusively eukaryotic ribosomes. For this purpose, each Latin name is now accompanied by a brief description in English. e.g. "parasitic eukaryote *Leishmania donovani*" instead of "*Leishmania donovani*" in the initial version.

I. 322-324. "For example, studies of *E. coli* under various growth conditions have shown that ribosomal proteins bL32, bL33, bL34, bS6, bS21 and bS20 can occasionally occur in at least 1.2 copies per ribosome." In the references added two-dimensional gel electrophoresis was used, a method that is not exactly quantitative. 20% variation does not definitely reflect the real stoichiometry. Please look at more recent papers.

We have deleted that statement and added additional references to more up-to-date studies of ribosome content using qMS (Refs. 67-69).

I. 358-360. "Particularly, we demonstrate that ribosomes do not have a strictly defined stoichiometry, where each protein is present in a single copy." This has been observed repeatedly, references would be nice.

We have addressed this comment by deleting the last section of our discussion.

Finally, in multiple places examples: line 6, 289, and 291. The words heterogenous and heterogeneous is used numerous times with apparently same meaning. While the words are similar in their meaning, they are not the same. A quick search revealed these differences in their meanings in the field of medicine and radiology:

- o Heterogeneous refers to a structure with dissimilar components or elements, appearing irregular or variegated.
- o Heterogenous refers to a structure having a foreign origin.

Thank you very much for this correction. We have now corrected the text by replacing "Heterogenous" with "Heterogeneous".

Reviewer #2 (Remarks to the Author):

Thank you.

Reviewer #3 (Remarks to the Author):

The authors present compelling structural evidence demonstrating that in *Psychrobacter urativorans*, compositional ribosome heterogeneity associated with the ribosomal protein bS20 does not result in functional specificity among distinct ribosomal pools. Using high-resolution cryo-EM, they identified two ribosomal populations, each containing either one or two copies of bS20. This finding was corroborated through in situ cryo-electron tomography (cryo-ET) subtomogram averaging, eliminating the possibility that the observed structural variability arose from the purification process. Furthermore, cryo-ET analysis enabled precise estimation of the localization and orientation of each ribosomal species within the cellular environment. Importantly, no significant differences were observed in translational states, subcellular localization, or polysome engagement between these ribosomal populations.

The study is well-executed, with clear and logical presentation of the results. I consider this work a robust contribution to the understanding of structural ribosomal heterogeneity and its lack of direct functional specificity. The experimental design and conclusions are sound, and I recommend the manuscript for publication after minor revisions.

We thank the reviewer for these kind remarks.

Minor Points for Consideration:

1. In the figures and text, the ribosomal “exit tunnel” is inconsistently referred to as the “tunnel exit.” I recommend standardizing this terminology.

We agree and have correct this to “exit tunnel”.

2. The captions for Figures 1 and 2 include concluding statements that seem more appropriate for the Results and Discussion sections. Specifically, sentences beginning with “Overall, this figure...” should be removed to avoid redundancy.

These sentences are now removed.

3. Supplementary Figures S1 and S2 are not referenced in the first section of the Results. Please ensure these figures are cited appropriately in the text.

This is now corrected.

Reviewer #4 (Remarks to the Author):

The manuscript of Helena-Bueno and co-workers addresses heterogeneity of translating ribosomes in prokaryote cells upon cold shock treatment.

This is not correct, and we apologise that our manuscript was not clear enough in this regard. While it is true that our **Fig. 1a,b** shows ribosomes isolated from cold-shocked cells, the rest of our data (**Fig. 1c,d** and **Figs. 2-3**) was all derived from *actively growing bacteria*, as stated in the results text:

*“To test whether bs20 binds to the large subunit of ribosomes constantly or only in response to stress, and to exclude its binding as an artifact of ribosome isolation, we determined the structure of P. urativorans ribosomes using cryo-electron tomography of intact, **non-stressed, exponentially growing** P. urativorans cells thinned by cryo-Focused Ion Beam (FIB) milling. “*

To resolve this confusion and make sure we articulate our results clearly, we have additionally:

- Included additional data showing a growth curve to demonstrate that bacterial cells transferred to cryo-EM grids were still exponentially growing at the time of plunge-vitrification (new **Supplementary Fig. S5**)
- Clearly stated in the figure captions that the data correspond to exponentially growing cell cultures.
- Expanded our results section to better emphasise the fact that all of the ribosomes in our cryo-ET dataset were associated with ligands of active translation - i.e. tRNAs, EF-Tu and EF-G. None were observed with vacant P sites or bound to known hibernation factors Balon and RaiA (**Supplementary Fig. S6**, formerly S3).

Ribosome heterogeneity within cells and its outcome in cell fate has been a well debated subject, and evidence of this physical heterogeneity has increased with state-of-the-art mass spectrometry and Cryo-EM. No longer is the ribosome simply viewed as a blindly translating machine, but has gained importance as a central player in the regulation of gene expression. The full extent of this ribosome heterogeneity and how it influences cell fate through translation is still poorly understood.

We appreciate the reviewer’s enthusiasm for the subject matter.

Here the authors seek to investigate the heterogeneity/segregation of translating ribosomes in Psychrobacter urativorans, specifically concerning the small ribosomal subunit protein bs20, (which is also able to bind to large ribosomal subunit) using Cryo Electro tomography (Cryo-ET) of FIB-milled cells. Although the topic is very exciting and the technique used to address this is state of the art, overall, the manuscript appears preliminary and the data provided are mostly in-silico analysis.

We thank the reviewer for highlighting the exciting nature of the work and our state-of-the-art technical approach. However, we respectfully disagree with the assessment of our work as “preliminary”. We would also like to challenge use of “*in silico*” - a term traditionally reserved for computer modelling and/or simulations. In the text, we conventionally use “*in*

situ” to mean structural studies of molecules in cells (i.e. within the cytoplasm rather than in test tubes). These “*in situ*” studies are data-driven experimental science.

Authors lack a control of cells grown in normal conditions,

We apologise for the confusion here. Our **Fig. 1c,d** and **Figs. 2-3** show only cells grown under optimal conditions. We have now emphasised this point as explained above.

and do not provide any biochemical data or literature complementing their Cryo-ET driven conclusions

To address this point, we now provide an additional Supplementary Figure (**Supplementary Fig. S5**) illustrating the active growth of our cultures used in the analysis, reproduced below for convenience.

Figure S5 | Growth curve of actively growing *P. urativorans* at physiological conditions. *P. urativorans* was cultured at 19 °C and constant shaking. For OD₆₀₀ measurements, 1 ml samples were taken every hour for 12 h in total. Samples for freezing were taken at an OD₆₀₀ of 0.58.

Such as in-vitro translation with purified polysomes to support that polysomes bearing one or two copies of bs20 translate at similar rates,

To address the reviewers concern regarding apparent translation rates, we now provide an additional analysis of pairwise distributions between all possible combinations of ribosomes with either 1x or 2x copies of bs20 (Figure 3e).

We thank the reviewer for the suggested biochemical experiments. Currently, these are not feasible because we have been unable to biochemically isolate polysomes or ribosome subunits of defined composition (regarding 1x vs. 2x bs20 copies) in a robust way. This is possibly due to the propensity of bs20 to dissociate from the large subunit during sucrose gradient ultracentrifugation, as previously demonstrated for this protein (ref 46 in the original manuscript version). Further, it is impossible to distinguish by e.g. western blot of polysome profiles, as antibodies will recognise bs20 equally well irrespective of whether this protein has originated from a binding site in the large or small subunit (the latter of which all ribosomes have). These difficulties are precisely why the “*in situ*” approach of structurally analysing ribosomes in the cytoplasm of intact cells is particularly valuable here.

or ribosome profiling to assess global polysome populations in cold shock conditions.

We have previously demonstrated that within 30 minutes of cold-shock, polysomes disappear and all ribosomes become associated with hibernation factors Balon and RaiA. We do not see any evidence of these factors in our cryo-ET datasets, consistent with actively growing cells. We now state this explicitly in the Results of our revised manuscript and more clearly highlight the absence of inactive ribosome classes in Figure S6 (formerly Figure S3, reproduced below for convenience):

Figure S6 | Cryo-ET data processing scheme for *P. urativorans*. Tilt frame stacks were pre-processed using MotionCor2 and Warp and binned tomograms were reconstructed in Warp. Ribosomal particles were localized using template matching implemented in PyTom. Reconstructed subtomograms were aligned in RELION and subjected to 3D classification. The resulting cryo-EM reconstruction was then used as input for local tilt-series refinement in M. Subsequently, optimized subtomograms were reconstructed and classified for quality and rotational state using a mask covering the 30S subunit. To identify translational states, rotated and unrotated ribosome populations were separately classified using a focus mask covering the factor binding site and the tRNA in A and P sites. To assess bS20 and Trigger Factor association, subtomograms were classified with focus masks for the bS20 and Trigger Factor binding sites on the 50S subunit.

Their conclusions are solely based on in-silico analysis of their Cryo-ET lamella of flash frozen bacteria grown in cold shock inducing conditions.

We apologise for the confusion. As mentioned above, none of our work is based on cryo-ET analysis of bacteria grown under cold shock-inducing conditions. All of our cryo-ET studies were performed exclusively using exponentially growing cultures prepared for these analyses.

Major points

1. The manuscript gives the reader a strong impression that bs20 binding to the 50S is something novel, however, it has been previously shown by several independent studies performed in various bacteria. Only in the discussion sections do the authors mention and cite that bs20 was previously known to bind to the 50S subunit. Hence, the introduction and results sections should better reflect this point.

To address this comment, we moved the corresponding section of our discussion to the introduction.

2. Do the 14 FIB-milled lamella represent different sections of the flash frozen bacteria? Or were bacteria always FIB-milled in the same region? If bacteria were milled in the same section, how can the authors be sure that there is no ribosome segregation?

FIB milling generated approximately central sections of arbitrarily oriented spherical cells, so no bias in terms of covered/imaged area is expected. This allows us to draw conclusions about the distribution of both ribosome populations throughout the entire cell.

3. Related to point 2. The authors should provide a Cryo-ET analysis of bacteria grown at normal temperatures in order to demonstrate that the Ribosome population is majorly formed by 1xbs20.

We apologise for the confusion. This is what the manuscript already shows.

Alternatively, could the authors show evidences of this control experiment with purified polysomes and Cryo-EM SPA? If there are no differences between physiological temperatures and cold shock, then the notion of ribosome heterogeneity is somehow misapplied here.

Please see our response above.

4. Authors state that Polysomes bearing 1 or 2 copies of bs20, have no significant cellular segregation (assuming point 2 has been addressed properly), and their different translation rates are not altered. While the techniques used in this paper fairly support the mixed polysomes, it does not substantially support that these ribosomes translate at the same rates simply based on distances between frozen ribosomes.

In these experiments, actively growing cells (new **Supplementary Fig. S5**) were snap-vitrified, so the “frozen ribosomes” that the reviewer refers to represent the natural distribution of all ribosomes in the cell, on the mRNAs that they happen to be translating at the point they were snap-vitrified. We know that the vast majority of these ribosomes are actively engaged in the elongation cycle, as judged by classification for tRNAs, rotation state and translational GTPases (**Fig. 3, Supplementary Fig. S6, formerly Supplementary Fig. S3**). No ribosome classes corresponding to inactive states were detected (**Supplementary Fig. S6, formerly Supplementary Fig. S3**).

If ribosomes bearing one or two copies of bS20 move at significantly different rates, it follows that we would expect to see collisions on polysomes containing both types of ribosomes. For example, if 2x bS20 ribosomes were to translate slower, we would expect to find 1x bS20 ribosomes accumulating behind 2x bS20 ribosomes - thereby changing the distributions for pairwise inter-ribosome distances on all polysomes in the cell. Indeed, ribosome queuing (disome and trisome formation) has been demonstrated for RQC substrates and at frameshift sites in bacteria and eukaryotes (55-59). However, we do not observe these collisions, or any statistically significant differences between inter-ribosome distances.

We believe that this analysis is more sensitive (and physiologically relevant) than any biochemical *in vitro* assay, that would necessarily be an average of multiple ribosomes of unknown composition, rather than an analysis of known annotated 1x bS20 and 2x bS20 individual ribosomes.

Other biochemical assays should be provided, or the text should be altered to fully contextualise the reader that the paper addresses this issue solely from an in-silico point of view.

We agree with the reviewer that our statements concerning elongation rate are an inference rather than kinetic data. We now better explain our reasoning (see response to point 4) and explicitly acknowledge the limitations of our analyses in the revised discussion.

5. No polysome profiling is shown (e.g. sucrose gradient) to demonstrate the global level of protein translation occurring in cells grown in cold shock or normal conditions. Is the ratio between polysomes and monosomes maintained? Do heavy/long polysomes still form?

Please see our response to point 1 – this study was not a comparison of cold-shock vs. normal conditions. We have previously demonstrated that within 30 minutes of cold-shock, polysomes disappear and the majority of ribosomes associate with RaiA and Balon hibernation factors (see Ref. 49, Figure 1a). We see no evidence of these hibernation factors here in our cryo-ET studies (**Supplementary Fig. S6**, formerly **Supplementary Fig. S3**).

6. Related to point 5. If possible, a Western Blot analysis of α -bS20 protein on 50S and 30S fractions originating from dissociated polysomes would also help prove the protein binding on the 50S. For instance, using puromycin in growing cells or high salt concentrations leads to 50S and 30S dissociation, which can then be purified by sucrose gradient, and fractions can be analysed by WB.

We thank the reviewer for this comment. However, the second bS20 binding site on the large subunit is lower affinity and more prone to dissociation during e.g. sucrose gradient ultracentrifugation (which was notably not used in our fast and mild *in vitro* sample preparation method, see Ref. 49). This problem is related to the historical ambiguity of whether bS20 was observed or not in different large subunit preparations (see Refs. 34-40). Therefore, we do not believe that this would be a more informative approach than our *in situ* structural analyses.

Nevertheless, to support our assignment of bS20 on the 50S subunit, we used ModelAngelo to build a protein backbone model and derive the most likely sequence of the resolved protein in a completely unbiased manner, using only the amino acid side chain features resolved for the putative bS20 density segment. Consistent with our assignment, ModelAngelo identified this segment of cryo-EM map as corresponding bS20. We now include this in our revised Methods and new **Supplementary Fig. S4** (reproduced below for convenience)

Fig S4. ModelAngelo results confirm the identity of previously unassigned density in the large ribosomal subunit of *P. urativorans* ribosomes as ribosomal protein bS20. (a) Workflow of automated model building using ModelAngelo to identify the masked density shown in the large ribosomal subunit of *P. urativorans* ribosome. (b) ModelAngelo output model docked in cryo-EM maps filtered by local resolution of 70S *P. urativorans* ribosomes shown in two orthogonal views. Close-up view shows the overall fit of segments of bS20 built using ModelAngelo into the large subunit of *P. urativorans* ribosomes. (c) Comparison of the previously determined bS20 structure in the small subunit (SSU) to ModelAngelo output model in the large subunit (LSU) shows the structural similarity between these models consistent with RMSD values calculated for individual helices (C α -atom RMSD α 1 = 0.901 Å, RMSD α 2 = 0.636 Å, RMSD α 3 = 0.403 Å). (d) Sequence alignment shows conserved residues of bS20 from *P. urativorans* and output sequence from ModelAngelo modelled helices.

7. Related to point 5. In a flash frozen sample, how reliably can monosomes close together be differentiated from polysomes? Can the authors track mRNA in their tomograms? This could either challenge and/or partially support that the authors “protein translation” results are not biased by sudden changes in ribosome/polysome total populations (fig 3 d and S.fig 6).

In cryo-ET, individual mRNA molecules cannot be resolved. It is therefore a standard approach in cryo-ET (Refs. 53, 54 in the revised manuscript) to computationally infer whether adjacent ribosomes are part of the same polysome or just come into random proximity by analysing their relative orientations. If they are on different mRNA molecules, they will adopt rotational poses that are unconstrained (i.e. all relative orientations are equally likely). If they are on the same mRNA molecule, the mRNA exit site of the leading ribosome will be more aligned towards the mRNA entry site of the trailing ribosome. We now describe this in more detail in the methods section and extended the results section:

“In recent cryo-ET studies of ribosomes, the native structure of polysomes was inferred based on the analyses of relative orientations of neighbouring ribosomes (Fig3c) (53,54).”

8. Related to point 5 – It would be interesting to provide biochemical data e.g cell-free translation systems prepared from normal or cold shock grown bacteria, and compare translation rates

Please see response to point 4 and 5 above.

9. In figure 3 panel c, it could be interesting to highlight bS20 bound to the 50S, to understand whether it is at the interface between ribosomes and somehow helps or impede translation process.

We thank the reviewer for this suggestion - the protein is now highlighted. bS20 bound to the 50S subunit is typically located away from the inter-ribosomal interface of polysomes.

10. Authors constructed a polyalanine based secondary structure in the bs20 (50S) map region, then searched in alpha fold seek for similar folds, yielding a list with top candidates representing bs20 protein from different bacteria. The authors further support this data with Mass Spec data analysis. Nevertheless, does the fold seek search outputs other proteins which could fit in this density?

No, bS20 was the only significant hit in our model organism. We now stated this point better in our revised Materials and Methods and Table S1.

Also, given the importance of this aspect for the whole paper, a figure with more detailed information should be provided to convince the reader that the bs20 density (50S side) confidently fits bs20 side chains.

We thank the reviewer for this suggestion. We have included an additional figure panel to show this in **Supplementary Fig. S3** (reproduced below for convenience) as well as a new **Supplementary Fig. S4** to present ModelAngelo-based identification of bS20 in the 50S subunit (see above).

Figure S3 | Cryo-EM density (blue mesh) and atomic models (yellow sticks) for bS20 $\alpha 1$, $\alpha 2$ and $\alpha 3$ helices on the large subunit. Density is contoured at 2.7 – 2.3 σ .

11. In figure S4a., authors compare local resolution plotted on Cryo-ET maps of ribosomes bearing 1 or 2 copies of bs20. In panel a 1xbs20 ribosomes globally point to a higher overall flexibility of solvent exposed surfaces, especially the RNA helix that in close proximity with bs20 binding site (the figure is small so it is hard to interpret correctly), as compared to 2xbs20 ribosomes. Can the authors comment on this? Would this also not affect the accurate determination of bs20 density on the 50S side, compromising the population distributions suggested by the authors?

Following the relative abundance of ribosome populations in the cytoplasm, the number of subtomograms in the 1xbs20 population (~12k particles) is significantly lower than in the 2xbs20 population (~25k particles). This leads to an overall lower resolution in the 1xbs20 population (8.2 Å vs 7.8 Å), which is also reflected in the lower local resolution of the solvent exposed surfaces in the 1xbs20 population. Nevertheless, resolutions in the range of 8 Å are sufficient to unambiguously identify presence or absence of alpha helical proteins - this is illustrated in **Fig. 2a**.

Minor points:

Figure S1 is mentioned in the text before Figure S3;

We have corrected the order of Supplementary Figures in the revised manuscript.

In Figure 1 “(OM)” is duplicated;

We have corrected the figure labelling.

Add resolution to relevant figures;

We have added resolution estimates to the relevant figures.

In Figure 3 consider adding in the subtitles the pymol script used for polysome detection;

We now reference the individual scripts in the methods section.

In figure 4 consider Changing “Zoomed-in” by “close-up view”.

We have adjusted the text as suggested.

I assume both models are from Cryo-ET reconstructions? If so, add this info in legend.

We assume the reviewer refers to figure 4. If so, no, these models are from the single-particle reconstructions. We have now clarified this in the figure 4 legend.

In Figure S4 consider circling the region where bs20 (on the 50S side) is located.

We have adjusted the figure as suggested.

In Figure S5, consider colouring maps for easier interpretation (e.g. 50S and 30S)

We have adjusted the figure as suggested.

Reviewer #5:

Thank you.